# Genome-Wide Identification of the *GhANN* Gene Family and Functional Validation of *GhANN11* and *GhANN4* under Abiotic Stress

**DOI:** 10.3390/ijms25031877

**Published:** 2024-02-04

**Authors:** Jin Luo, Meili Li, Jisheng Ju, Han Hai, Wei Wei, Pingjie Ling, Dandan Li, Junji Su, Xianliang Zhang, Caixiang Wang

**Affiliations:** 1State Key Laboratory of Aridland Crop Science, College of Life Science and Technology, Gansu Agricultural University, Lanzhou 730070, China; 18992209045@163.com (J.L.); limeili369@163.com (M.L.); 18893855307@163.com (J.J.); 13653802626@163.com (H.H.); 17865557657@163.com (W.W.); linger12012021@163.com (P.L.); m15293180496@163.com (D.L.); sujj@gsau.edu.cn (J.S.); 2Institute of Cotton Research, State Key Laboratory of Cotton Biology, Chinese Academy of Agricultural Sciences (CAAS), Anyang 455000, China

**Keywords:** *annexin* genes, *Gossypium hirsutum*, abiotic stresses, virus-induced gene silencing (VIGS), gene expression

## Abstract

Annexins (ANNs) are a structurally conserved protein family present in almost all plants. In the present study, 27 *GhANNs* were identified in cotton and were unevenly distributed across 14 chromosomes. Transcriptome data and RT-qPCR results revealed that multiple *GhANNs* respond to at least two abiotic stresses. Similarly, the expression levels of *GhANN4* and *GhANN11* were significantly upregulated under heat, cold, and drought stress. Using virus-induced gene silencing (VIGS), functional characterization of *GhANN4* and *GhANN11* revealed that, compared with those of the controls, the leaf wilting of *GhANN4*-silenced plants was more obvious, and the activities of catalase (CAT), peroxidase (POD), and superoxide dismutase (SOD) were lower under NaCl and PEG stress. Moreover, the expression of stress marker genes (*GhCBL3*, *GhDREB2A*, *GhDREB2C*, *GhPP2C*, *GhRD20-2*, *GhCIPK6*, *GhNHX1*, *GhRD20-1*, *GhSOS1*, *GhSOS2* and *GhSnRK2.6*) was significantly downregulated in *GhANN4*-silenced plants after stress. Under cold stress, the growth of the *GHANN11*-silenced plants was significantly weaker than that of the control plants, and the activities of POD, SOD, and CAT were also lower. However, compared with those of the control, the elasticity and orthostatic activity of the *GhANN11*-silenced plants were greater; the POD, SOD, and CAT activities were higher; and the *GhDREB2C*, *GhHSP*, and *GhSOS2* expression levels were greater under heat stress. These results suggest that different *GhANN* family members respond differently to different types of abiotic stress.

## 1. Introduction

A large family of proteins called annexins (ANNs), which bind to membrane phospholipids in a calcium-dependent manner, has been found in both animals and plants [1]. Since the initial cloning of ANNs from human cells [2], more than a thousand ANN members have been discovered. ANN protein family members are divided into five classes: vertebrate ANNs belong to class A; invertebrate ANNs are classified as class B; ANNs of fungi and some single-celled eukaryotes belong to class C; plant ANNs belong to class D; and prokaryotic ANNs are classified as class E [3]. Although plant ANNs were first studied earlier than animal ANNs were, many valuable advances have been made in plant ANNs in recent years. In 1989, plant annexin was first isolated from isolated tomato cells [4]. To date, accumulating scientific data indicate that plant *annexins* are broadly dispersed across various plant tissues, with distinct functions assigned to each subtype. It was found that *annexins* were significantly expressed in the root hairs and fibers of some plants. *PvAnn1* is highly expressed at the site of symbiosis between root hairs and *Rhizobium tropici*, and a decrease in *PvAnn1* expression leads to the shortening of root hairs in kidney beans [5]. On the other hand, *AnnSp2*, which was cloned from naturally drought-resistant plants (*Solanum pennellii*), was expressed in all organs of tomato plants but was expressed at lower levels in roots and more highly expressed in leaves and flowers [6]. Previous studies have shown that ANN protein levels increase in response to red light during the elongation stage and that *p35* results in a greater level of staining in secretory cell types, such as outer root cap cells. *p35* immunostaining is primarily found in secretory cells, such as outer root cap cells and root hair cells, in pea plants [7,8,9]. These results imply that tip-directed exocytosis processes, including secretion and the generation of new cell walls, are significantly influenced by ANN proteins. In addition, *ANN* genes play a potential role in the reproductive stage. Studies have shown that *TaAnn10* is specifically expressed in anthers but is not expressed at low temperatures in heat-sensitive male sterile lines [10]. Similar results have been observed in *Arabidopsis* [11]. Subsequent studies revealed that increased *AtAnn5* expression levels are specific to *Arabidopsis* anthers and that inhibition of *AtAnn5* expression leads to decreased pollen grain size, increased pollen mortality, and delayed pollen tube growth [11]. On the other hand, abnormal developmental stages from the vegetative to the reproductive stage, as well as during embryogenesis, are also observed in *AtAnn5* RNA interference plants [11]. Additionally, the expression levels of *FaAnn5a* and *FaAnn8* are higher during strawberry fruit ripening, which is promoted by exogenous ABA and inhibited by IAA [12]. According to previous research, TtGS5, a serine carboxypeptidase-like protein found in *Triticum timopheevii*, interacts with TtAnnD1 to control the size and weight of premature grains [13]. Furthermore, the expression levels of *annexins* were much greater at night than during the day in *Mimosa pulvinus*, suggesting that *MpANNs* may play a unique role in circadian rhythm control [14].

In addition to its vital role in growth and development, *Annexin* is crucial for plant responses to biotic stress. *Annexins* promote signal transduction by perceiving signals on the membrane, enabling plants to defend themselves against biotic stress. Previous research has demonstrated that salicylic acid increases the expression level of *AtAnn1*, which is possibly related to the function of proteins related to biotic stress [15]. Additionally, it was discovered that the *ann1ann2* double mutant was more susceptible to *Botrytis cinerea* [16]. *AtANN8* inhibits cell death and reduces powdery mildew resistance mediated by RPW8.1 [17]. After infection with *Pseudomonas syringae*, the expression level of *NtAnn12* was markedly elevated in tobacco [18]. Studies have also shown that *ANN* expression is induced by bacterial wilt, root-knot nematode, and thrips infestations in tomato and pepper plants [19]. Plants generate proteins known as pathogenesis-related (PR) proteins in response to pathogen invasion. Upon pathogen invasion, ANN induces the expression of several PR proteins [20,21,22]. *CkANN* can increase peroxidase activity, induce the expression of several PR protein genes, and further enhance resistance to *Fusarium oxysporum* in transgenic cotton lines [22]. *AnnAt1* and *AnnAt4* are targeted by MiMIF-2 in *Arabidopsis thaliana* (MiMIF-2 is an inhibitor of macrophage migration in root-knot nematodes). While plants with *AnnAt1* and *AnnAt4* knocked out become more susceptible to root-knot nematodes, plants that overexpress *AnnAt1* and *AnnAt4* exhibit increased resistance to these worms [23].

*Annexins* are crucial in plants for regulating abiotic stress, as they respond to biotic stress. Numerous studies have demonstrated that ANN proteins can protect plants against abiotic stress [20]. Elevated salinity is the most common abiotic stress on plants. Salt stress induces the cytoplasmic transport of AtAnn1 to membranes in *A. thaliana*, and *annAt1*/*annAt4* double mutants exhibit hypersensitivity to ABA signaling and osmotic stress [24]. Studies have also shown that *Ann1At* and *Ann4At* interact to regulate salt and drought tolerance in a Ca^2+^-dependent manner [20,25]. An increasing number of taxa have recently demonstrated annexin responses to drought and salt stress. An increasing number of studies conducted in the past several years have demonstrated that ANN proteins play a role in how plants react to stressors such as drought and salt. Under salt and drought stress, the degree of growth inhibition in *AnnSp2*-overexpressing tomato plants was lower than that in wild-type plants [6]. Studies have shown that overexpressing *PtAnnexin1* can enhance plant resilience to salt and drought stress in artificial poplar plants [26]. The ability of transgenic lines to withstand salt stress can be enhanced by overexpressing *AnnBj2* in tobacco [27,28]. Studies have shown that *GhANN1* can improve the drought and salt tolerance of upland cotton plants by controlling ion homeostasis, regulating the accumulation of ABA, and participating in the phenylpropanoid production pathway and that *GhAnn8b* can improve the salt tolerance of transgenic *Arabidopsis* lines by elevating Na^2+^ efflux [29,30,31]. *OsAnn3* actively controls rice drought stress resistance in an ABA-dependent manner, while *OsAnn10* may negatively regulate the osmotic stress response because *OsAnn10*-knockdown plants exhibit tolerance to osmotic stress [32,33]. Several studies have shown that cold stress can increase the accumulation of ANN in the wheat plasma membrane [34]. *AtAnn1* mutants exhibit decreased freezing resistance during cold-induced [Ca^2+^]_cyt_ elevation [35]. A further experiment supported this conclusion, showing that phytoannexin positively regulates cold stress [35]. On the other hand, the *OsAnn3* loss-of-function mutant shows resistance to cold [36]. *ZmANN33* and *ZmANN35* were found to play active roles in plant recovery from cooling damage in maize experiments [37]. Earlier studies revealed that the expression of *NnANN1* increased the heat resistance of *Arabidopsis* seeds [38]. Rapid elevation of [Ca^2+^]_cyt_ in plants is considered an important step in response to heat stress [39]. The *ann1*/*ann2* double mutants are more sensitive to heat shock treatment, and both ANNs redundantly control heat shock tolerance in *Arabidopsis* [39]. Additional research has shown that MYB30 depends on *AtAnn1* and *AtAnn4* to control the responses to heat stress and oxidative stress [40]. It has been hypothesized that the function of ANN in plant heat tolerance is related to peroxidation regulation [41]. By promoting catalase (CAT) and superoxide dismutase (SOD) activities, which regulate REDOX homeostasis and H_2_O_2_ concentration, the overexpression of *OsANN1* improves the development of rice plants under abiotic stress [41]. 

Cotton is an important cash crop in addition to being a raw material used in the textile industry. The cotton *annexin* gene was shown to be significantly expressed during the cotton fiber differentiation extension phase, suggesting that *annexin* is involved in cell elongation [42,43]. Additional research has shown that *GhAnn2* controls Ca^2+^ flow signaling, which in turn controls fiber formation [43]. Because abiotic stress significantly reduces cotton production and fiber quality, it is crucial to leverage key genes to increase the abiotic stress resistance of cotton varieties. The systematic study of *ANNs* in cotton is not only helpful for studying the resistance mechanism of *ANN* family genes but also has the potential to facilitate the use of *ANN* family genes to improve the stress resistance of cotton varieties. In this work, *ANN* gene family members were identified in four different *Gossypium* spp. using bioinformatics approaches. Evolutionary linkages, expression patterns, chromosomal locations, and gene structural characteristics were determined. To filter potential *GhANNs*, RNA-seq and qRT-PCR data were obtained. We discovered that at least three stress treatments increased the expression levels of *GhANN4* and *GhANN11*. To investigate their impact on the abiotic stress response of upland cotton plants, *GhANN4* and *GhANN11* were silenced by VIGS to examine their roles in salt, drought, and low-temperature and high-temperature responses. In addition to providing good genetic resources for the production and use of *GhANNs* to produce resistant cotton varieties, this work will further our understanding of the importance of the *ANN* gene family in the cotton abiotic stress response. 

## 2. Results

### 2.1. Identification and Physicochemical Properties of ANN Gene Family Members in Four Gossypium *spp*.

A total of 97 *ANN* candidate genes were extracted from *Gossypium arboreum*, *Gossypium raimondii*, *Gossypium hirsutum*, and *Gossypium barbadense*. Fourteen *GaANNs*, fourteen *GrANNs*, twenty-seven *GhANNs*, and twenty-nine *GbANNs* were identified after excluding candidate genes with significantly deficient ANN domains and those without ANN domains (Table 1 and Appendix A), and they were named based on their chromosomal positions. More than 80% of the ANN proteins from the four *Gossypium* spp. contained 304 to 328 amino acids. The results of isoelectric point analysis showed that ANN proteins are either acidic or basic and that each type was present in roughly equal proportions in the four *Gossypium* spp. (*G. hirsutum*: thirteen acidic, fourteen basic; *G. barbadense*: thirteen acidic, sixteen basic; *G. arboreum*: seven acidic, seven basic; *G. raimondii*: seven acidic, seven basic). Among the four *Gossypium* spp., the instability indices of most GhANN proteins were less than 40, while those of the GbANN, GaANN, and GrANN proteins were all greater than 40. Only the GhANN proteins were relatively stable [44]. Approximately 67% of the ANN proteins were predicted to be present in the cytoplasm of the four *Gossypium* spp. (Table 1 and Appendix A). 

### 2.2. Phylogenetic Study of the ANN Gene Family Members

To thoroughly investigate the evolutionary link of *ANNs* in different species, one hundred and seventy-four ANN proteins from 10 species were concatenated to construct an evolutionary tree (fourteen GaANNs, fourteen GrANNs, twenty-seven GhANNs, twenty-nine GbANNs, ten OsANNs, twenty-five TaANNs, twelve PtANNs, twelve ZmANNs, twenty-three GmANNs, and eight AtANNs) in this study. Based on bootstrap values (=1000), the findings showed that six main clades (A–F) consisting of 174 ANNs were clustered together (Figure 1). Clade A had the lowest distribution (*n* = 19, 10.92%), while Clade B had the highest distribution (*n* = 40, 22.99%). The GhANNs were mainly clustered in clade B, clade E, and clade F. In addition, the *ANNs* from the four cotton species were more concentrated in each group.

### 2.3. Distribution of Chromosomes, Intergenomic Interactions, and Collinear Relationships within the GhANN Family

Chromosome distribution analysis of the 27 *GhANN* genes revealed that they were scattered on 14 chromosomes with a certain distribution preference (Figure 2A). To investigate duplicates within the *GhANN* family, we performed a genomic collinearity analysis in MCScanX. Eight segmentally and two tandemly duplicated gene pairs were found throughout the upland cotton genome (Figure 2B). This finding suggests that segmental duplication events made a greater contribution than tandem duplication events in the *GhANN* family. The majority of the *GhANNs* were found to be homologous on the chromosomes of *G. arboreum* and *G. raimondii*, and certain *GhANNs* correlated with multiple homologous genes in both species (Figure 2C). Like those in *G. arboreum* and *G. raimondii*, these homologous genes in *G. hirsutum* species are likely expressed in comparable tissues or cell types and have comparable functions. The *Ka*:*Ks* ratios of the *GhANN* family members were lower than 0.82 (Appendix A), which indicated that purifying selection was highly important for the evolution of *GhANNs*.

### 2.4. Conserved Polypeptide Motifs, Functional Domains, and Gene Structure of the ANN Family

The conserved polypeptide motifs of 27 GhANN proteins were assessed using MEME software (https://meme-suite.org/meme/, accessed on 30 April 2023) [45]. The results demonstrated that all the GhANNs had at least motifs 2 and 4, and GhANN13 had all the motifs (Figure 3A). Only GhANN5 and GhANN11 did not have annexin domains, while GhANN5 had only annexin superfamily domains, and ANX domains only existed in GhANN4, GhANN9, GhANN19, GhANN20, and GhANN21 (Figure 3B). We examined the genomic DNA sequences of *GhANNs* from *G. hirsutum* to determine the numbers of exons and introns and the arrangement of each gene. The exon counts ranged from four to six for the *GhANNs* (Figure 3C). Additionally, the evolutionary tree revealed that closely related *GhANN* family members had more conserved motifs and gene structures. We speculated that the functions of these *GhANN* family members with conserved motifs could be similar.

### 2.5. Examination of Cis-Acting Elements within the GhANN Gene Family

The findings demonstrated that the TATA-box and CAAT-box, two essential components that primarily guarantee the precision and effectiveness of gene transcription initiation, were present in the promoters of the 27 *GhANN* genes. Additionally, components related to growth and development, light response, stress response, and hormone response were found in the promoters of these genes (Figure 4). With eight components, the most prevalent *cis*-acting element associated with growth and development was the CAT-box. The *GhANN* promoters had hormone response elements, including those related to salicylic acid (TCA-element), jasmonic acid (CGTCA-motif, TGACG-motif), auxin (TGA-element), gibberellin (P-box, TATC-box), and ABRE. The most prevalent of these cis-acting elements were ABREs. Many stress response-related cis-acting elements, such as anaerobic induction elements (AREs), cold response elements (LTRs), defensive stress elements (TC-rich repeats), and drought response elements (MBS), were also found in the GhANN promoter region, among which the number of AREs was the greatest (57). Among the light response cis-acting elements, we identified many Box 4, G-box, and GT1-motif elements.

### 2.6. GhANN Gene Expression Patterns under Different Abiotic Stresses

To explore the expression profiles of the *GhANN* genes, the expression levels of the genes were characterized under different abiotic stresses using FPKM values to create heatmaps. Genes that showed a twofold difference in expression were classified as differentially expressed in comparison to controls (CK). Some *GhANN* genes were highly expressed after treatment with various tissues, NaCl and PEG, or cold and heat (Figure 5 and Appendix A). For example, the transcript levels of *GhANN4*, *GhANN9*, *GhANN10*, *GhANN11*, *GhANN14*, *GhANN23*, and *GhANN25* were higher than those of CK (0 h) after NaCl and PEG treatment (Figure 5A), and the transcript levels of *GhANN4*, *GhANN9*, *GhANN11*, *GhANN14*, and *GhANN23* were higher than those of CK (0 h) after cold and heat treatment (Figure 5B). The transcript levels of *GhANN4*, *GhANN9*, *GhANN19*, *GhANN21*, *GhANN23*, and *GhANN25* were high in all tissues (Appendix A). Taken together, these results indicate that abiotic stress has a significant impact on the expression of several *GhANNs* in upland cotton.

### 2.7. Validation of GhANN Gene Expression Patterns under Abiotic Stress

To further investigate the role of the *ANN* genes, eight *GhANN* genes were selected for qRT-PCR analysis. Overall, the eight candidate *GhANN* genes were more responsive to PEG than to NaCl, cold, or heat (Figure 6). Excluding *GhANN11* and *GhANN27*, the other six *GhANN* genes responded strongly to NaCl stress, and the expression levels of these six genes exhibited a double hump-like trend, with two peaks occurring at 1 h and 6 h after NaCl stress (Figure 6). Under PEG stress, other than *GhANN15*, the expression levels of the other seven *GhANNs* tested after PEG stress were hundreds of times greater than those in the control (0 h) (Figure 6). All eight *GhANNs* tested responded slightly to cold stress, and the highest expression levels were found after 6 or 24 h of cold stress treatment (Figure 6). Under heat stress, the peak response of *GhANN11* occurred within 1 h of stress, and the peak expression of the other genes, excluding *GhANN23* and *GhANN25*, appeared after 24 h of stress (Figure 6). Taken together, these findings indicate that *GhANNs* might participate in cotton resistance to abiotic stresses. 

### 2.8. Silencing of GhANN4 Attenuated Drought and Salt Tolerance in Upland Cotton

We found that the expression of *GhANN4* was induced by abiotic stress. We used virus-induced gene silencing (VIGS) to silence the *GhANN4* gene in upland cotton to study the function of the gene in the response to salt and drought stress. After the photobleaching of leaves infected with TRV:*GhCLA*, the efficiency of *GhANN4* silencing was determined via qRT-PCR analysis in the TRV:00 and TRV:*GhANN4* plants (Figure 7A and Figure 8A). Compared to that in the control plants (TRV:00), the expression level of *GhANN4* was 50% lower in the TRV:*GhANN4* plants (Figure 7B and Figure 8B). At the two-true leaf stage, we treated control plants and *GhANN4*-silenced plants with 15% PEG and 400 mM NaCl (Figure 7C and Figure 8C). We discovered that the leaves of the *GhANN4*-silenced plants withered more than the leaves of the control plants and that the *GhANN4*-silenced plants also exhibited more leaf shrinkage after 35 days of PEG and NaCl stress (Figure 7D and Figure 8D).

POD, SOD, CAT, and MDA activity levels were determined in the leaves of TRV:00 and TRV:*GhANN4* plants treated with PEG and NaCl stress (Figure 7E and Figure 8E). The results showed that the MDA activity in the TRV:*GhANN4* plants was not significantly greater than that in the TRV:00 plants and that the POD, SOD, and CAT activities in the TRV:00 and TRV:*GhANN4* plants decreased to varying degrees following PEG treatment (Figure 7E), with the TRV:00 plants exhibiting considerably greater POD activity than the TRV:*GhANN4* plants under NaCl stress. When TRV:*GhANN4* plants were subjected to NaCl stress, SOD activity and CAT activity decreased significantly, while MDA activity increased to varying degrees (Figure 8E).

The expression levels of six drought stress tolerance-related marker genes (*GhCBL3*, *GhDREB2A*, *GhDREB2C*, *GhPP2C*, *GhRD20-2*, and *GhRD29A*) were measured, and we discovered that the expression of these genes was greater in the TRV:*GhANN4* plants than in the control plants without PEG treatment; on the other hand, the expression of these marker genes was lower in the TRV:*GhANN4* plants than in the control plants after PEG treatment (Figure 7F). After measuring the expression levels of six marker genes associated with salt stress tolerance (*GhCIPK6*, *GhNHX1*, *GhRD20-1*, *GhSOS1*, *GhSOS2*, and *GhSnRK2.6*), we found that these genes had higher expression levels in TRV:*GhANN4* plants than in control plants that did not receive NaCl treatment; on the other hand, the expression of these marker genes was lower in TRV:*GhANN4* plants (Figure 8F).

### 2.9. Effect of GhANN11 Gene Silencing on Cold Tolerance and Heat Resistance in Cotton

Based on transcriptome data and RT-qPCR data for *GhANNs* under abiotic stress, we found that cold and heat stress strongly induced *GhANN11* expression. To study the effect of *GhANN11* on cold and heat stress, we silenced *GhANN11* in upland cotton plants via VIGS. After the leaves of TRV:*GhCLA*-infected plants were photobleached, the efficiency of *GhANN11* silencing was determined in TRV:00 and TRV:*GhANN11* plants via qRT-PCR analysis (Figure 9A and Figure 10A). Compared with that in the control plants (TRV:00), the transcript level of *GhANN11* in the TRV:*GhANN11* plants was reduced by 50% (Figure 9C and Figure 10C). At the two-true leaf stage, control plants and *GhANN11*-silenced plants were treated at 10 °C and 42 °C (Figure 9B and Figure 10B). We found that the leaves of the *GhANN11*-silenced plants were more withered than those of the control plants after 10 days of cold stress, while the leaves of the *GhANN11*-silenced plants exhibited greater elasticity and orthostatic properties than those of the control plants after 6 days of heat stress (Figure 9D and Figure 10D).

Under cold and heat stress, POD, SOD, and CAT activities were determined in the leaves of TRV:00 and TRV:*GhANN11* plants (Figure 9E and Figure 10E). The findings demonstrated that, under cold stress, the POD, SOD, and CAT activities of the TRV:*GhANN11* plants were significantly lower than those of the TRV:00 plants (Figure 9E), but under heat stress, the POD, SOD, and CAT activities of the TRV:*GhANN11* plants were significantly greater than those of the TRV:00 plants (Figure 10E).

We determined the expression levels of four cold tolerance-related marker genes (*GhDREB2A*, *GhRD20-1*, *GhRD29A*, and *GhWRKY33*) and found that their expression levels were greater in TRV:*GhANN11* plants than in control plants without cold stress. However, under cold stress, the expression levels of these marker genes were lower in the TRV:*GhANN11* plants than in the control plants (Figure 9F). The expression levels of three marker genes associated with high-temperature stress resistance (*GhDREB2C*, *GhHSP*, and *GhSOS2*) were measured, and we discovered that the expression of these genes was greater in TRV:*GhANN11* plants than in control plants that did not receive high-temperature treatment. On the other hand, the expression of these marker genes in the TRV:*GhANN11* plants significantly increased after high-temperature stress (Figure 10F).

## 3. Discussion

When examining the role and development of a gene family, accurate and comprehensive identification is essential. Several plants, including *Arabidopsis* [46], wheat [10], rice [47], and *Glycine max* [48], have been shown to harbor *ANN* genes thus far. The family has eight to twenty-five members. Like in the majority of angiosperms, 27 *GhANNs* were found in *Gossypium hirsutum*. A total of 174 genes from 10 species were chosen for the construction of a phylogenetic tree to determine the evolutionary links between *ANNs*. The *ANNs* were then categorized into six clusters, or clades A–F. Monocot and eudicot *ANN* genes were present in all the clades, indicating that the six branches were differentiated before monocot and eudicot separation.

Gene duplication events, which produce new functional genes and promote species development, occurred in all the species. Upland cotton is a heterotetraploid cotton species composed of the A_t_ subgenome and D_t_ subgenome [49]. As expected, orthologs of almost all the *GhANNs* were found in *G. arboreum* and *G. raimondii*, which is consistent with the findings in closely related species. We identified 13 and 14 GhANN genes in the A_t_ subgenome and D_t_ subgenome, respectively, and most of the genes exhibited one-to-one collinear relationships. Only *GhANN10* was found in the D_t_ genome but not in the A_t_ subgenome and may have been lost during evolution. Most of the genes originated from segmental duplication (eighteen pairs), and a few genes originated from tandem duplication (two pairs) (Appendix A). These findings indicate that segmental duplication is the main driving force of *GhANN* expansion and that tandem duplication also plays a role in this process.

Numerous stress and development mechanisms have been linked to *ANN* genes. *TdAnn6* can strengthen a plant’s defense against salt and drought stress [50]. The resilience of poplar plants to salt stress and drought can be enhanced by overexpressing *PtAnnexin1* [26]. The germination rate of transgenic cotton plants overexpressing the *GhAnn1* gene was much greater than that of wild-type plants, in which the former had a longer root system and more vigorous development under drought and salt stress [29]. These findings are in line with our discovery of cis-acting elements in the upstream promoter region of *GhANNs* that are connected to plant hormones, growth, and development. Transcriptome data from *G. hirsutum* showed that different *GhANNs* exhibited different expression patterns under different abiotic stresses, suggesting that the functions of these *GhANNs* may be different. For example, the expression of several *GhANNs* was not induced under salt and drought stress (*GhANN7*, *GhANN12*, *GhANN13*, *GhANN20*), and the expression of several *GhANNs* was not induced under high- or low-temperature stress (*GhANN7*, *GhANN13*, *GhANN20*). The expression patterns of eight genes (*GhANN4*, *GhANN9*, *GhANN11*, *GhANN14*, *GhANN15*, *GhANN23*, *GhANN25*, and *GhANN27*) were further confirmed by qRT-PCR, and the results were highly compatible with the RNA-seq data, demonstrating the validity of the expression data. We found that the expression level of *GhANN4* changed significantly under salt stress and drought stress at different times, while the expression level of *GhANN11* increased significantly under heat stress and cold stress at different times. These findings suggest that *GhANN4* and *GhANN11* may have important biological functions under these conditions. To better understand its function, we next investigated their biological functions’ response to salt, drought, heat, and cold stress, respectively.

A growing body of evidence suggests that the ability to withstand abiotic stress is influenced by antioxidant enzyme activity [51]. Reactive oxygen species (ROS) production may increase as a result of abiotic stress, which may reduce a plant’s ability to tolerate it. Plants have developed intricate ROS clearance mechanisms to evade the negative effects of ROS [52]. To further investigate the role of *GhANN4* and *GhANN11* in the cotton abiotic stress response, we constructed TRV:*GhANN4*- and TRV:*GhANN11*-silencing vectors. NaCl and PEG were applied to the TRV:00 and TRV:*GhANN4*-silenced plants, while high and low temperatures were used to treat the TRV:00 and TRV:*GhANN11*-silenced plants. The elimination of reactive oxygen species is facilitated by the antioxidant system, an essential defense mechanism. The main components of the system include CAT, POD, and SOD, and indicators of plant resistance can be better understood by examining their activities [53]. We discovered that when silenced plants were subjected to salt, drought, and low-temperature stress, their POD, SOD, and CAT activities greatly decreased; however, when they were subjected to high-temperature stress, their activities dramatically increased. Increased activity of antioxidant enzymes contributes to plants’ capacity to eliminate reactive oxygen species, which lowers membrane lipid peroxidation and preserves the integrity of the membrane structure. We found that the leaves of the target gene-silenced plants wilted significantly under salt, drought, and low-temperature stress, while the leaves of the control plants were not affected, and the leaves of the control plants wilted more significantly than those of the target gene-silenced plants under high-temperature treatment. This aligns with the findings of our earlier studies. An important indicator of the degree of damage to the plasma membrane and membrane lipid peroxidation is the MDA concentration. In the present study, under conditions of salt and drought stress, the MDA content of the silenced plants increased more considerably than that of the control plants, suggesting that the silenced plants suffered additional oxidative damage.

Previous research has demonstrated that the overexpression of TFs may control the expression of genes that respond to stress and improve plant resistance to a range of stressors. Overexpression of *TdAnn6* in wheat enhanced tolerance to high salt by regulating stress-related genes [50]. The expression of the salt tolerance gene *MsAnn2* in alfalfa was enhanced under salt stress [54]. To better understand the resistance of *GhANN* to abiotic stress, we examined the expression of marker genes associated with abiotic stress, including *GhCBL3*, *GhDREB2A*, *GhDREB2C*, *GhRD20-2*, *GhSOS1*, and *GhNHX1* [55,56,57,58,59,60,61,62,63,64,65,66,67,68], in TRV:00 and TRV:*GhANN4* plants. Our results showed that the expression levels of *GhCBL3*, *GhDREB2A*, *GhDREB2C*, *GhPP2C*, and *GhRD20-2* in *GhANN4*-silenced plants were significantly downregulated under drought stress, and the expression levels of *GhCIRK6*, *GhNHX1*, *GhRD20-1*, *GhSOS1*, *GhSOS2*, and *GhSNRK2.6* were also significantly downregulated under salt stress, suggesting that silencing *GhANN4* may weaken the ROS clearance ability of *G. hirsutum* by reducing the expression of these genes, thus reducing drought and salt tolerance. Similarly, the expression levels of *GhDREB2A*, *GhRD20-1*, *GhRD29A*, and *GhWRKY33* in *GhANN11*-silenced plants significantly decreased under low-temperature stress, while the expression levels of *GhDREB2C*, *GhHSP*, and *GhSOS2* increased in the *GhANN11*-silenced plants under high-temperature stress. These findings suggested that *GhANN11* protects against temperature extremes by upregulating the expression of *GhDREB2A*, *GhRD20-1*, *GhRD29A*, and *GhWRKY33* and downregulating the expression of *GhDREB2C*, *GhHSP*, and *GhSOS2*. On the basis of the above results, we speculated that *GhANN4* positively regulates the tolerance of upland cotton to salt and drought and that *GhANN11* positively regulates the tolerance of upland cotton to low temperatures and negatively regulates the tolerance of upland cotton to high temperatures. 

## 4. Materials and Methods

### 4.1. Identification and Analysis of ANN Gene Family Members

The ANN protein sequence from *Arabidopsis* was used as a seed sequence to construct a hidden Markov model from the Pfam database (http://pfam.xfam.org/, accessed on 18 April 2023). With the use of HMMER 3.0 software, the threshold was set to e < 1.0 × 10^−5^, and the HMM file was used to search the four *Gossypium* species protein databases from CottonFGD (http://www.cottonfgd.org, accessed on 18 April 2023) to obtain candidate protein sequences. Using the online software platforms PfamScan (https://www.ebi.ac.uk/Tools/pfa/pfamscan/, accessed on 18 April 2023) and SMART (http://smart.embl-heidelberg.de/, accessed on 18 April 2023), all the candidate gene sequences were assessed. Using the online tool ExPASy (https://web.expasy.org/, accessed on 19 April 2023), the physicochemical properties of the GhANN protein, including the number of amino acids, molecular weight, and isoelectric point (pI), were analyzed. Subcellular localization prediction analysis was performed on the WoLF PSORT (https://wolfpsort.hgc.jp/, accessed on 19 April 2023) subcellular localization website.

### 4.2. Phylogenetic Tree Construction, Chromosome Mapping, and Collinearity Analysis of GhANN Family Genes

We performed multiple sequence alignment of the GhANNs using the ClustalW tool. The phylogenetic tree was subsequently generated with MEGA 13 software using the neighbor method with 1000 bootstrap replicates and subsequently submitted to iTOL for visualization. Using MapInspect software (http://mapinspect.software.informer.com, accessed on 28 April 2023), the positions of the *GhANN* family genes were mapped on the chromosome. For collinearity analysis, the GhANN, GaANN, and GrANN protein sequences were aligned with each other using the Basic Local Alignment Search Tool (BLAST) with a cutoff E value (<10^−5^). The abovementioned BLASTP results were evaluated by the MCScanX (https://mybiosoftware.com/tag/mcscanx, accessed on 29 April 2023) tool of TBtools (v1.09876) to generate collinearity blocks covering the whole genome. Collinear pairs of GhANN, GaANN, and GrANN family proteins were extracted to construct a collinearity map using TBtools software. Subsequently, the *Ka*/*Ks* values of the *GhANN* genes were determined with TBtools software.

### 4.3. Examination of the GhANN Family Gene Structure and Conserved Motifs in Gossypium hirsutum

The online application MEME [69] was used to examine the conserved polypeptide motifs found in the ANN proteins of *G. hirsutum*. The settings were adjusted to a maximum motif number of 10 and ideal group widths between 6 and 50. The Pfam v33.1-18271 PSMM database was used to identify conserved functional domains via NCBI CDD software (https://www.ncbi.nlm.nih.gov/Structure/cdd/wrpsb.cgi, accessed on 30 April 2023), with the other settings remaining at the default values [69]. For the *GhANN* genes, the gene structures and intron counts were determined using the Gene Structure Display Server 2.0 program (http://gsds.cbi.pku.edu.cn/, accessed on 30 April 2023) [70]. The program TBtools was used to visualize the results [71].

### 4.4. Prediction of Cis-Acting Regulatory Elements within GhANN Promoters

Using the PlantCARE promoter analysis program, the 2 kb upstream sequences of the *GhANN* family members were obtained from the CottonFGD database and examined to identify cis-acting regulatory elements in the GhANN promoter regions [72].

### 4.5. Analysis of the Gene Expression Characteristics of the GhANN Family in Gossypium hirsutum

RNA-seq data from *G. hirsutum* plants were obtained from Zhejiang University (http://cotton.zju.edu.cn/, accessed on 1 May 2023) [73]. Transcriptome datasets derived from plants subjected to varying levels of heat, cold, drought, and salt stress were also obtained.

### 4.6. Real-time Fluorescence-Based Quantitative PCR (qRT-PCR) of Selected GhANNs

XinshiK25 seeds were soaked in sterile water for approximately 24 h, exposed to light, and subsequently placed in a homemade water culture box for germination. Cotton plants with similar growth were selected and transferred to an improved Hoagland nutrient solution for cultivation in a light culture room. When the plants reached the four-leaf stage, they were subjected to stress treatment with 200 mmol·L^−1^ NaCl and 15% PEG-6000 solution, with three biological replicates for each treatment. Some XinshiK25 plants were subjected to cold (12 °C) and heat (42 °C) stress. Leaves were sampled from all treatments at 0, 1, 3, 6, 12, and 24 h. 

A polysaccharide polyphenol total RNA extraction kit (Tiangen Biochemical Technology Co., Ltd., Beijing, China) was used to extract total RNA from the collected samples, and the purity, content, and integrity were determined with an ultramicro concentration detector and agarose gel electrophoresis. Reverse transcription was performed using the FastKing gDNA Dispatching RT SuperMix Kit (Tiangen Biochemical Technology Co., Ltd.). qPCR primers were designed using the NCBI Primer BLAST tool (https://www.ncbi.nlm.nih.gov/tools/primer-blast/, accessed on 1 May 2023), with product fragment sizes ranging from 130 to 170 bp (Appendix A), and synthesized by Shanghai Shenggong Biotechnology Service Co., Ltd. (Shanghai, China). The reaction system was prepared using a Talent qPCR PreMix (SYBR Green) fluorescence quantitative reagent kit (Tiangen Biochemical Technology Co., Ltd.): 10 µL Talent qPCR PreMix, 2 µL cDNA, 2.4 µL each of forward/reverse primers, and 3.2 µL RNase-free H_2_O. Using a Roche instrument for RT-PCR, the amplification program was performed at 95 °C for 180 s, 95 °C for 5 s, 60 °C for 15 s, and 40 cycles. Three technical replicates were performed for each sample. Using the 2^−ΔΔCT^ method, the relative expression level of each gene was calculated. The gene expression maps were drawn using Origin software. (2022)

### 4.7. VIGS of GhANN4 and GhANN11 in G. hirsutum

To create the TRV:*GhANN4* and TRV:*GhANN11* constructs, segments of *GhANN4* (312 bp) and *GhANN11* (414 bp) were inserted into the *Eco*RI and *Kpn*I restriction sites of the TRV-based (pYL156) vector. Appendix A displays all the primers used for vector construction. The *Agrobacterium tumefaciens* strain GV3101 was transfected with the helper vectors pYL192, TRV:*GhANN4*, TRV:*GhANN11*, TRV:00, and TRV:*GhCLA1*. The strains harboring pYL192 and TRV:00, TRV:*GhCLA1*, TRV:*GhANN4*, or TRV:*GhANN11* were subsequently combined at a 1:1 ratio and incubated for three hours at 28 °C. After infiltrating the cotyledons of 8-day-old cotton plants, the mixed A. tumefaciens strain solution was administered to produce *GhANN4*- and *GhANN11*-silenced cotton plants (TRV:*GhANN4* and TRV:*GhANN11*), as well as negative (TRV:00) and positive (TRV:*GhCLA1*) control plants. The target gene was expressed in both *GhANN4*- and *GhANN11*-silenced cotton plants and control plants when the TRV:*GhCLA1* plants displayed an albino phenotype.

### 4.8. Salt, Drought, Cold, and Heat Resistance Stress Treatment

Silenced plants and negative control plants were subjected to salt, drought, cold, and heat stress. The roots of both control and target *GhANN4*-silenced plants were irrigated with 400 mM NaCl as salt stress up to 35 d. The roots of both control and target *GhANN4*-silenced plants were irrigated with 15% PEG6000 as drought stress up to 35 d. For cold tolerance evaluation, plants were placed at 10 °C for 10 d. For heat resistance evaluation, plants were placed at 42 °C for 6 d. 

### 4.9. SOD, POD, and CAT Activity and MDA Content Determination

The leaves of untreated and treated control plants and the *GhANN4*- and *GhANN11*-silenced plants were ground in a mortar with liquid nitrogen into powder, and 3 mL of phosphate buffer (pH 7.8) was added to the mixture, which was subsequently ground for 2 min until homogenized. Afterward, 2 mL of buffer was used to rinse the mortar, and the homogenizing liquid was transferred to a centrifuge tube. After centrifugation for 20 min, the supernatant was collected and stored at 4 °C for the determination of antioxidant enzyme activity and MDA content. The nitroblue tetrazole photoreduction method was used to measure SOD activity, the guaiacol colorimetric approach was used to measure POD activity, and the colorimetric method was used to measure the MDA concentration in plant tissues [74]. The approach of [75,76] was used to assess CAT activity. Throughout the measurement period, three technical replicates were used for each sample.

## 5. Conclusions

Plant tolerance to stress is substantially influenced by the *ANN* gene family. In *G. hirsutum*, we found 27 *ANN* genes. Phylogenetic, gene structure, transcription pattern, protein motif, and subcellular localization analyses were performed. We discovered that following salt and drought treatment, *GhANN4* expression was markedly upregulated. Additionally, silencing *GhANN4* increased the sensitivity of upland cotton plants to PEG and salt stress. Under stress from high and low temperatures, *GhANN11* expression changed in distinct ways. This research identified great candidate genes for genetic engineering to increase cotton stress resistance in addition to providing fundamental data for research on the *GhANN* genes in cotton.

## Figures and Tables

**Figure 1 ijms-25-01877-f001:**
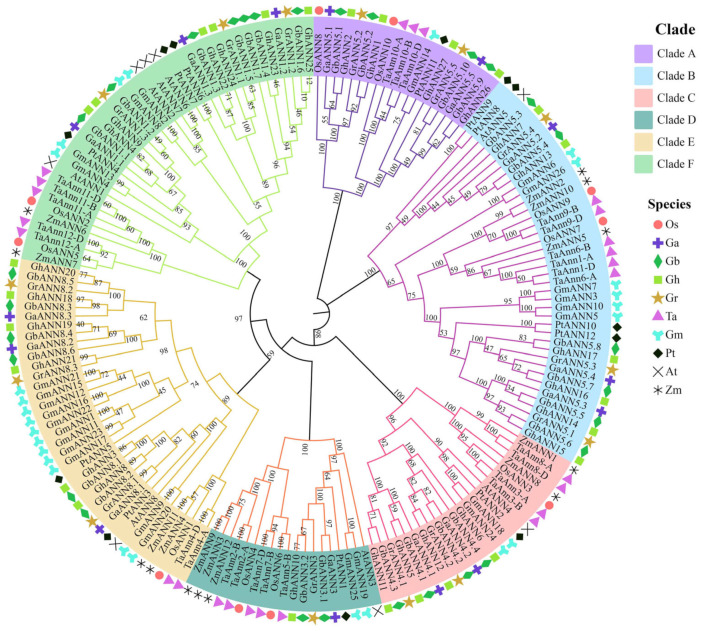
Phylogenetic tree of ANN orthologs from *G. hirsutum*, *G. raimondii*, *G. arboretum*, *Triticum aestivum*, *Oryza sativa*, *Populus*, *Zea mays*, *Glycine max*, and *A. thaliana*. Using 1000 bootstrap replicates, the neighbor method of MEGA 13 software was used to create the tree. The red dot represents Os, the purple cross represents Ga, the dark green rhombus represents Gb, the pale green square represents Gh, the khaki five-pointed star represents Gr, the lilac triangle represents Ta, the cyan tri-point shape represents Gm, the black diamond represents Pt, the black cross represents At, and the black asterisk represents Zm.

**Figure 2 ijms-25-01877-f002:**
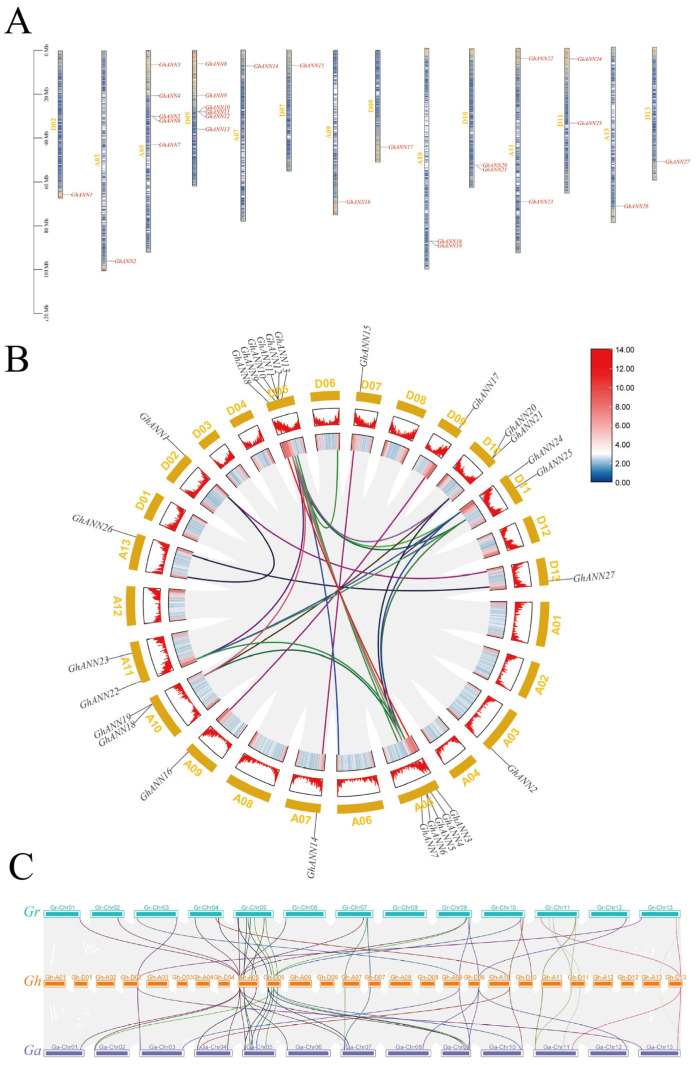
Chromosomal distribution, intergenomic relationships, and collinear relationships of the *GhANN* family. (**A**) The distribution of the 27 *GhANN* genes on the *Gossypium hirsutum* chromosomes. (**B**) Patterns of *GhANN* gene duplication in the genome of *Gossypium hirsutum*. (**C**) The genome-scale collinear link between the 27 *GhANN* genes and the *ANN* families of *Gossypium arboreum* and *Gossypium raimondii*.

**Figure 3 ijms-25-01877-f003:**
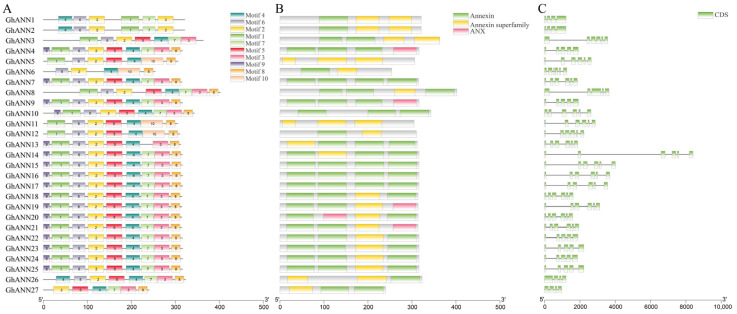
Structural analysis of the GhANNs. (**A**) Distribution of motifs in the 27 GhANNs. (**B**) Distribution of conserved functional domains in the 27 GhANNs. (**C**) Exon/intron structure of the *GhANN* genes.

**Figure 4 ijms-25-01877-f004:**
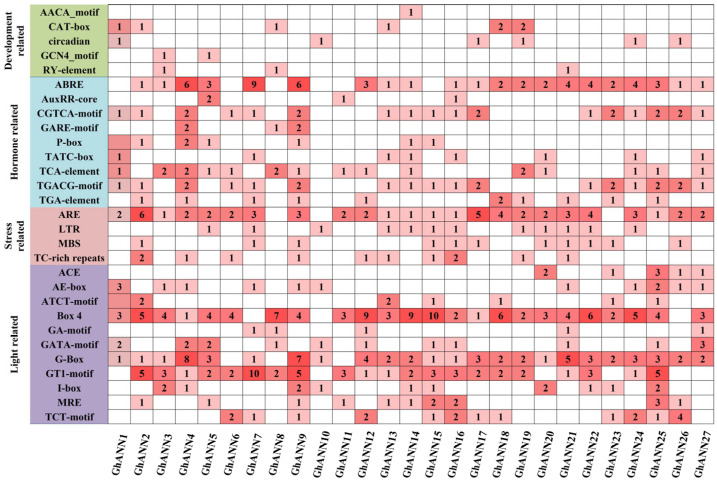
Cis-regulatory elements in the promoter regions of *GhANN* gene family members. The number in each cell represents the number of cis-acting elements present in each *GhANN* promoter region.

**Figure 5 ijms-25-01877-f005:**
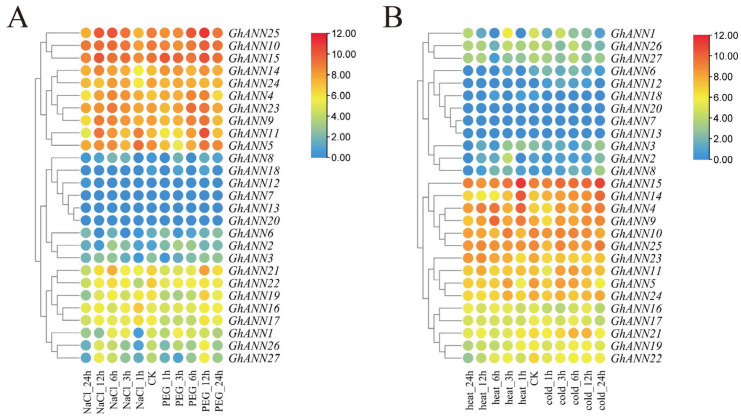
The *GhANN* gene expression profiles under different stresses. (**A**) *GhANN* gene expression patterns during PEG and NaCl stress. (**B**) *GhANN* gene expression patterns in response to heat and cold stress. High and low expression levels are represented by red and blue, respectively.

**Figure 6 ijms-25-01877-f006:**
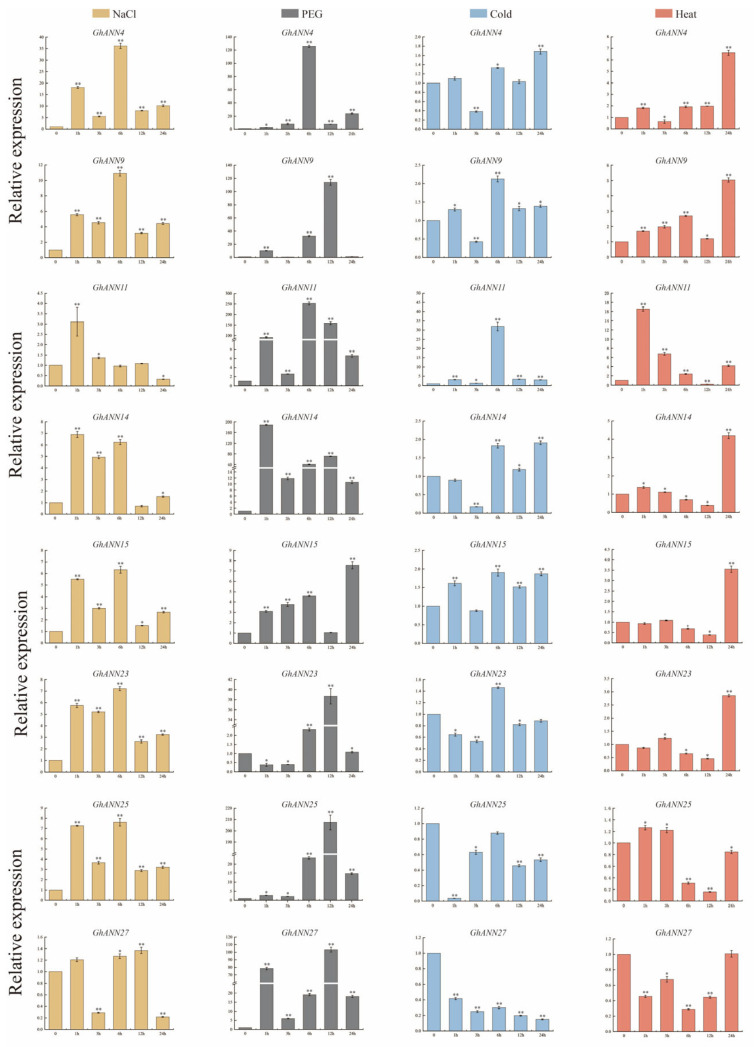
The expression levels of eight *GhANNs* under NaCl, PEG, cold, and heat stress. The data for three biological replicates are represented by error bars, which show the standard deviations. Yellow, gray, blue, and pink represent PEG, NaCl, cold, and heat stress, respectively. The asterisks indicate significant differences according to Student’s *t* test. *, *p* < 0.05; **, *p* < 0.01.

**Figure 7 ijms-25-01877-f007:**
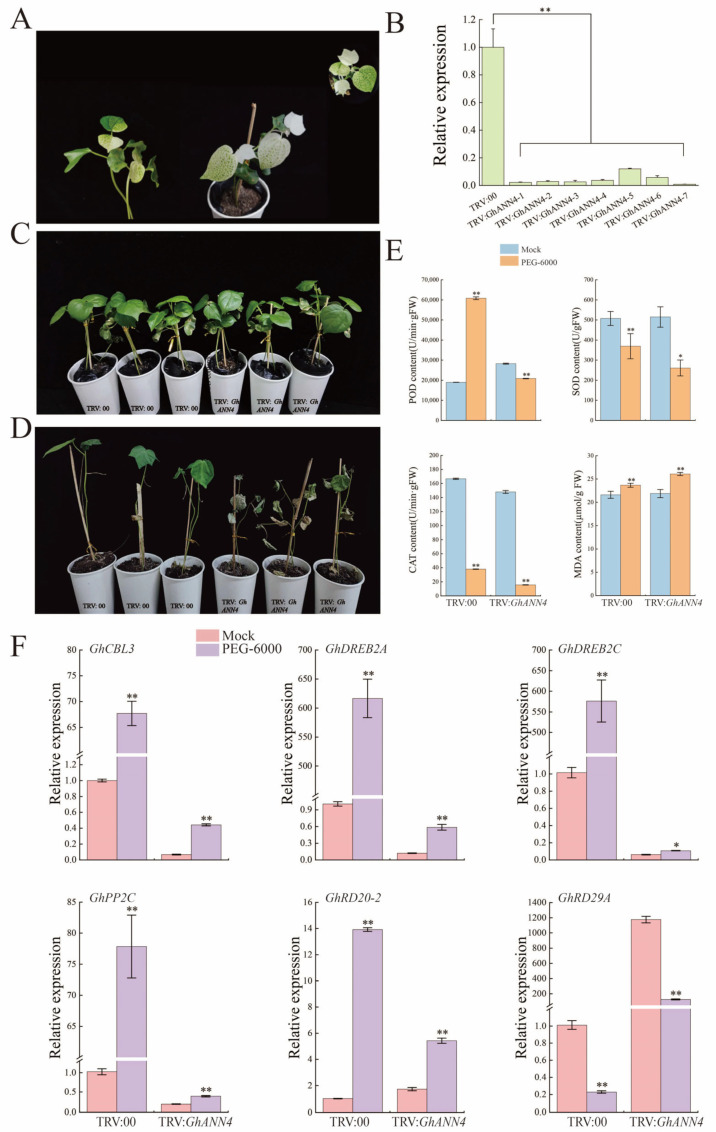
Silencing of *GhANN4* compromised cotton tolerance to PEG stress. (**A**) Positive control plants. (**B**) Quantitative RT-PCR analysis of *GhANN4* expression in TRV:00 plants and TRV:*GhANN4* plants. For the silenced plants, the expression of the target genes was half that of the control plants, and these plants were selected for stress treatment. *GhActin* (AY305733) was used as an internal control. SD denotes the standard deviation calculated from three independent experiments. (**C**) Cotton phenotypes of the control (TRV:00) and *GhANN4*-silenced (TRV:*GhANN4*) plants without stress treatment. (**D**) Cotton plant phenotypes under drought stress were controlled (TRV:00) or silenced by *GhANN4* (TRV:*GhANN4*). Photos were taken after 35 days of PEG treatment. (**E**) SOD activity, POD activity, CAT activity, and MDA content of *GhANN4*-silenced and control plants under normal conditions and PEG treatment. (**F**) Quantitative RT-PCR analysis of marker genes related to drought stress tolerance in silenced and control plants before and after PEG treatment. The asterisks indicate significant differences according to Student’s *t* test. *, *p* < 0.05; **, *p* < 0.01.

**Figure 8 ijms-25-01877-f008:**
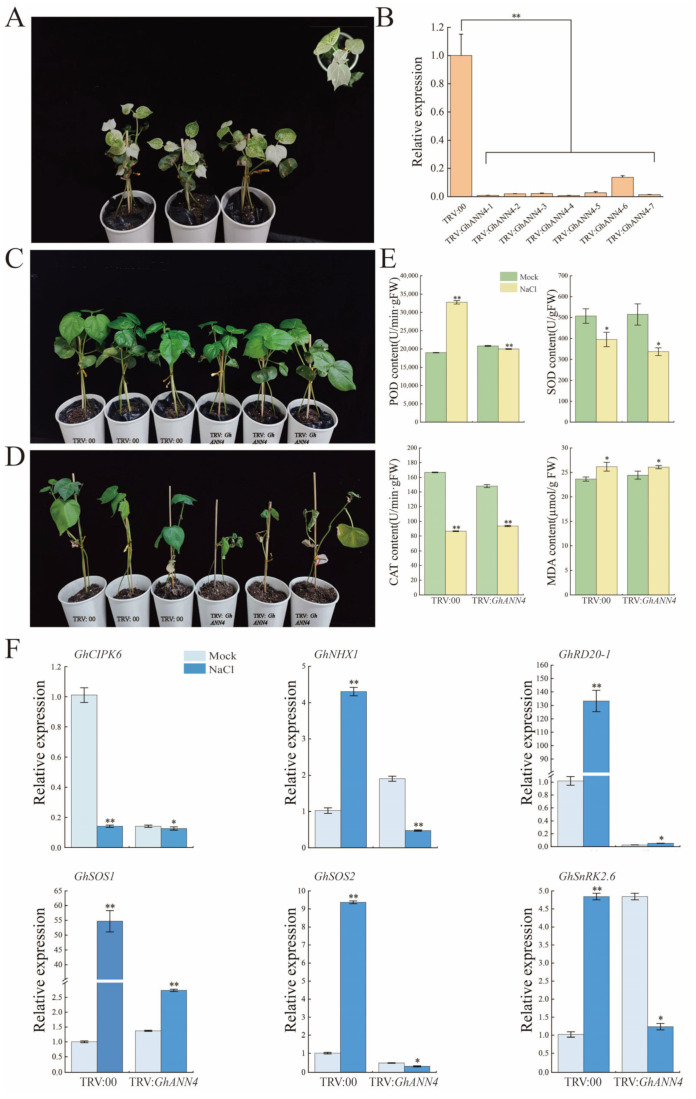
Silencing of *GhANN4* compromised cotton tolerance to NaCl stress. (**A**) Positive control plants. (**B**) Quantitative RT-PCR analysis of *GhANN4* expression in TRV:00 plants and TRV:*GhANN4* plants. Silenced plants in which the expression of the target genes was half that in the control plants were selected for stress treatment. *GhActin* (AY305733) was used as an internal control. SD denotes the standard deviation calculated from three independent experiments. (**C**) Cotton plant phenotypes were silenced by the control (TRV:00) or *GhANN4* (TRV:*GhANN4*) without stress treatment. (**D**) Cotton plant phenotypes under NaCl stress were controlled (TRV:00) or silenced by *GhANN4* (TRV:*GhANN4*). Photos were taken after 35 days of NaCl treatment. (**E**) SOD activity, POD activity, CAT activity, and MDA content of *GhANN4*-silenced cotton plants and controls under normal conditions and NaCl treatments. (**F**) Quantitative RT-PCR analysis of marker genes related to NaCl stress tolerance in silenced and control plants before and after NaCl treatment. The asterisks indicate significant differences according to Student’s *t* test. *, *p* < 0.05; **, *p* < 0.01.

**Figure 9 ijms-25-01877-f009:**
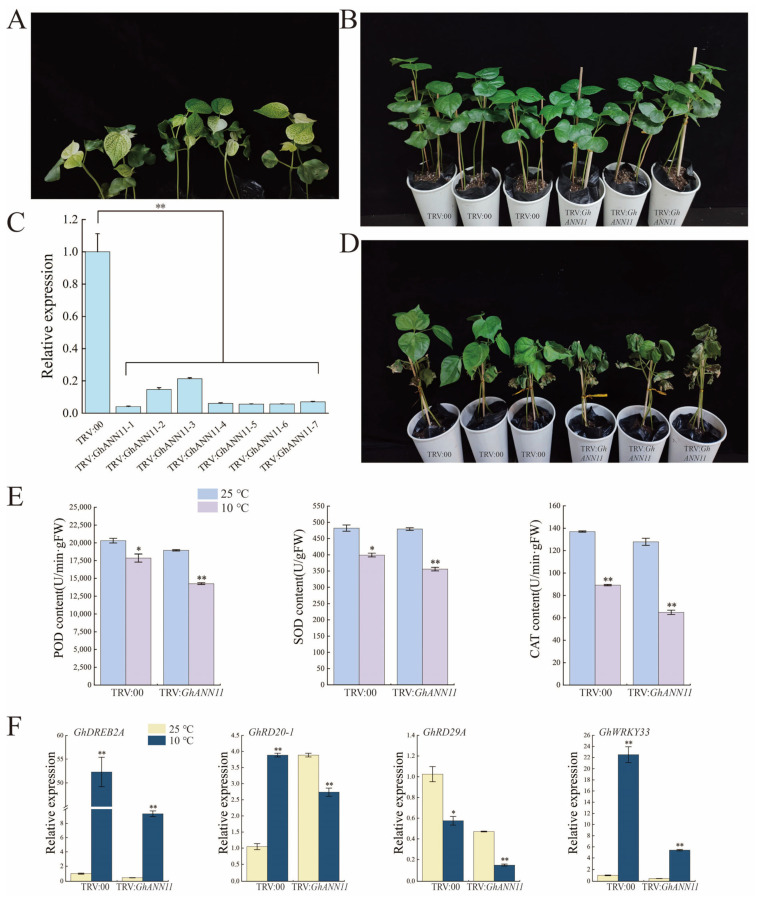
Silencing of *GhANN11* compromised cotton tolerance to cold stress. (**A**) Positive control plants. (**B**) Quantitative RT-PCR analysis of *GhANN11* expression in TRV:00 plants and TRV:*GhANN11* plants. For the silenced plants, the expression of the target genes was half that of the control plants, and these plants were selected for stress treatment. *GhActin* (AY305733) was used as an internal control. SD denotes the standard deviation calculated from three independent experiments. (**C**) Cotton plant phenotypes were silenced by the control (TRV:00) or *GhANN11* (TRV:*GhANN11*) without stress treatment. (**D**) Cotton plant phenotypes under cold stress in the control (TRV:00) or silenced by *GhANN11* (TRV:*GhANN11*). Photos were taken after 10 days of cold treatment. (**E**) SOD activity, POD activity, and CAT activity of GhANN11-silenced and control plants under normal conditions and cold treatments. (**F**) Quantitative RT-PCR analysis of marker genes related to cold stress tolerance in silenced and control plants before and after cold treatment. The asterisks indicate significant differences according to Student’s *t* test. *, *p* < 0.05; **, *p* < 0.01.

**Figure 10 ijms-25-01877-f010:**
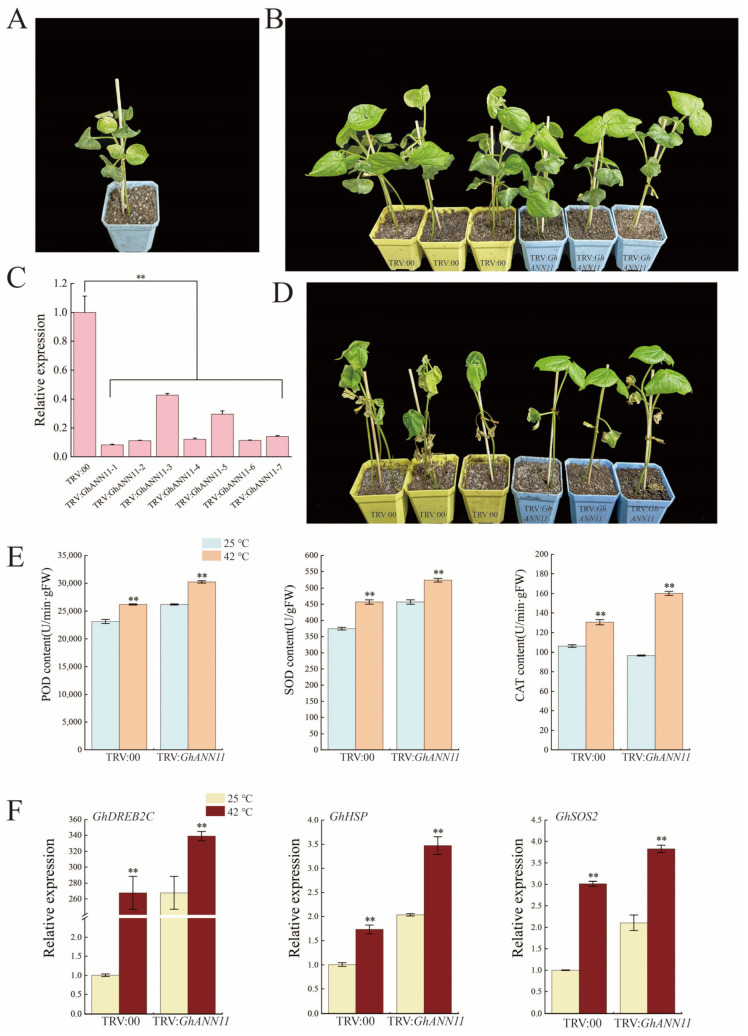
Silencing of *GhANN11* impairs cotton tolerance to heat stress. (**A**) Positive control plants. (**B**) Quantitative RT-PCR analysis of *GhANN11* expression in TRV:00 plants and TRV:*GhANN11* plants. Silenced plants in which the expression of target genes was half that in control plants were selected for stress treatment. *GhActin* (AY305733) was used as an internal control. SD denotes the standard deviation calculated from three independent experiments. (**C**) Cotton plant phenotypes were silenced by control (TRV:00) and *GhANN11* (TRV:*GhANN11*) without stress treatment. (**D**) Cotton plant phenotypes under heat stress were controlled (TRV:00) and silenced by *GhANN11* (TRV:*GhANN11*). Photos were taken after 6 days of heat treatment. (**E**) SOD activity, POD activity, and CAT activity of *GhANN11*-silenced cotton plants and controls under normal conditions and heat treatments. (**F**) Quantitative RT-PCR analysis of marker genes related to heat stress tolerance in silenced and control plants before and after heat treatment. The asterisks indicate significant differences according to Student’s *t* test. **, *p* < 0.01.

**Table 1 ijms-25-01877-t001:** Basic information on the *ANN* gene family in cotton.

Genes Name	Genes ID	Number of Amino Acids (aa)	Molecular Weight (kDa)	Theoretical pI	The Instability Index	Grand Average of Hydropathicity (GRAVY)	Subcellular Localization
*GhANN1*	*GH_D02G2140.1*	321	36.53	9.71	42.78	−0.307	nucleus
*GhANN2*	*GH_A03G1973.1*	321	36.37	9.75	42.37	−0.312	nucleus
*GhANN3*	*GH_A05G0751.1*	363	41.14	6.65	42.58	−0.425	Golgi apparatus
*GhANN4*	*GH_A05G2275.1*	316	35.97	6.39	35.10	−0.474	cytoplasm
*GhANN5*	*GH_A05G2824.1*	306	34.70	6.88	45.71	−0.395	cytoskeleton
*GhANN6*	*GH_A05G2825.1*	254	29.37	8.66	31.04	−0.558	cytoplasm
*GhANN7*	*GH_A05G3169.1*	315	35.56	9.23	36.60	−0.422	cytoplasm
*GhANN8*	*GH_D05G0748.1*	402	45.62	6.47	41.72	−0.31	plasma membrane
*GhANN9*	*GH_D05G2295.1*	316	36.00	6.34	33.42	−0.487	cytoplasm
*GhANN10*	*GH_D05G2841.1*	343	38.47	6.28	36.16	−0.33	peroxisome
*GhANN11*	*GH_D05G2842.1*	305	34.60	6.33	44.93	−0.372	cytoskeleton
*GhANN12*	*GH_D05G2843.1*	310	35.24	8.82	32.27	−0.496	cytoplasm
*GhANN13*	*GH_D05G3187.1*	312	35.22	9.1	38.53	−0.416	cytoplasm
*GhANN14*	*GH_A07G0681.1*	316	35.57	9.08	37.04	−0.324	cytoplasm
*GhANN15*	*GH_D07G0672.1*	316	35.45	9.1	36.77	−0.273	cytoplasm
*GhANN16*	*GH_A09G1894.1*	316	35.49	9.11	37.07	−0.304	chloroplast
*GhANN17*	*GH_D09G1846.1*	316	35.57	9.12	37.39	−0.327	chloroplast
*GhANN18*	*GH_A10G2000.1*	314	36.02	8.45	50.23	−0.624	cytoplasm
*GhANN19*	*GH_A10G2002.1*	314	35.83	6.78	38.79	−0.557	cytoplasm
*GhANN20*	*GH_D10G2102.1*	314	36.08	8.79	51.66	−0.615	cytoplasm
*GhANN21*	*GH_D10G2103.1*	314	35.84	6.74	40.63	−0.525	cytoplasm
*GhANN22*	*GH_A11G0554.1*	316	35.77	6.23	29.41	−0.429	cytoplasm
*GhANN23*	*GH_A11G2122.1*	316	36.01	6.19	36.18	−0.467	cytoplasm
*GhANN24*	*GH_D11G0581.1*	316	35.85	6.39	31.13	−0.451	cytoplasm
*GhANN25*	*GH_D11G2320.1*	316	36.06	6.19	36.66	−0.453	cytoplasm
*GhANN26*	*GH_A13G1903.1*	323	36.62	8.05	47.01	−0.373	cytoplasm
*GhANN27*	*GH_D13G1859.1*	240	27.19	7.17	35.31	−0.387	cytoskeleton

## Data Availability

All the data generated or analyzed during this study are included in this article and its Appendix A.

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
