# Peer review of "Genome-Wide Identification of the GhANN Gene Family and Functional Validation of GhANN11 and GhANN4 under Abiotic Stress"

_ijms, 2024, doi:10.3390/ijms25031877_

Round 1
Reviewer 1 Report
Comments and Suggestions for Authors
The article entitled "Knockdown of GhANN4 increases cotton sensitivity to PEG and salt stress, and knockdown of GhANN11 enhances cotton tolerance to heat stress" has analyzed the nitrogen utilization efficiency in rapeseed. The authors indentificaition of GhANN gene family and functional validation of GhANN1 and GhANN4. Some issues should be concerned.
1. The title is not suitable to contain the content of the whole manuscript. The VIGS is just for gene silencing instantaneous, not transgenic process. The results in this manuscript are manily on genome wide indentificaition of GhANN gene family and functional validation of GhANN11 and GhANN4 under abiotic stresses.
2. According to the authors' results, silencing of GhANN11 compromised cotton tolerance to cold stress. However, the title and abstract were not included this result.
3. How about Ka/Ks for evolution analysis of GhANN gene family in cotton or the four Gossypium species ? And the expression patterns of GhANN genes in different tissues?
4. Why the authors choose the two GhANN genes, GhANN11 and GhANN4 for functional validation? The authors should clarify this point and discussion.
5. How about the cis-acting elements of GhANN11 and GhANN4 with the functional validation for PEG, NaCl, cold and heat?
6. Figure 8 E, There is some doubt of the significant between mock and NaCl for POD (** P < 0.01), while SOD and MDA is just (* P < 0.05)?
7. The English language requires polish. For instance,
(1) The Abstract should concise, 11 stress marker genes for GhANN4-silenced plants , but none description of the marker genes for GhANN11 ?
(2) Line 248 “different stress” should be “different stresses”
Comments on the Quality of English Language
The English language requires polish.
Author Response
Thank you for the informative comments and helpful suggestions from you . We have studied the comments carefully and have made revisions. All the comments were valuable and provided important guidance for our research. The manuscript has been polished by English-speaking experts. The revisions made according to the comments from the reviewers have further improved our manuscript. We carefully revised the manuscript accordingly, and the changes are marked in yellow in the text. The responses are as follows:
Comments 1: The title is not suitable to contain the content of the whole manuscript. The VIGS is just for gene silencing instantaneous, not transgenic process. The results in this manuscript are manily on genome wide indentificaition of GhANN gene family and functional validation of GhANN11 and GhANN4 under abiotic stresses.
Response 1: Thank you for this constructive suggestion. “Knockdown of GhANN4 increases cotton sensitivity to PEG and salt stress, and knockdown of GhANN11 enhances cotton tolerance to heat stress” was changed to “Genome-wide identification of the GhANN gene family and functional validation of GhANN11 and GhANN4 under abiotic stress” on lines 2 to 4.
Comments 2: According to the authors' results, silencing of GhANN11 compromised cotton tolerance to cold stress. However, the title and abstract were not included this result.
Response 2: Thank you for the reminder. We changed the title according to comments 1 and added the result that GhANN11 silencing harms cold resistance in cotton plants in the abstract . Please see lines 25 to 27.
Comments 3: How about Ka/Ks for evolution analysis of GhANN gene family in cotton or the four Gossypium species ? And the expression patterns of GhANN genes in different tissues?
Response 3: We apologize for the lack of these two results. We have added the relevant results for upland cotton Ka/Ks in section 2.3 on lines 204 to 206 and Table S2, and added the relevant results analysis method in section 4.2 on lines 532 to 533. The Ka:Ks ratios of the GhANN family members were lower than 0.82, which indicated that purifying selection was highly important for the evolution of GhANNs. We have added the expression patterns of the upland cotton plants in the different tissues to Figure S1. The transcript levels of GhANN4, GhANN9, GhANN19, GhANN21, GhANN23, and GhANN25 were high in all the tissues. Please see Figure S1 and lines 259 to 260.
Comments 4: Why the authors choose the two GhANN genes, GhANN11 and GhANN4 for functional validation? The authors should clarify this point and discussion.
Response 4: Thank you for the helpful reminder. We have explained in the discussion why GhANN4 and GhANN11 were chosen for functional verification; please see lines 451 to 456.
Comments 5: How about the cis-acting elements of GhANN11 and GhANN4 with the functional validation for PEG, NaCl, cold and heat?
Response 5: This is a good question. There is a relationship between promoter cis-acting elements and gene function, and many genes that perform certain functions have specific cis-acting elements in their promoter region. The GhANN4 promoter is responsive to hormones and has a large number of AREs and ABREs, which play a certain role in the regulation of drought and salt stresses. The GhANN11 promoter is responsive to light and also contains AREs, which regulate the stress effect of plants. The plant response to photoperiod is a physiological and behavioral adaptation mechanism that can regulate plant growth, flowering, photosynthesis and other physiological and behavioral activities and can regulate plants under cold and heat stresses to a certain extent so that plants can adapt to environmental changes. GhANN promoters contain stress response elements and hormone response elements, which may be involved in the regulation of cotton expression in response to stress.
Comments 6: Figure 8 E, There is some doubt of the significant between mock and NaCl for POD (** P < 0.01), while SOD and MDA is just (* P < 0.05)?
Response 6: This is a good question. When cotton plants are stressed, a large amount of reactive oxygen species are produced. To resist this attack, POD, SOD, CAT and MDA in the cotton plant cooperate to clear oxygen free radicals. The total antioxidant enzyme activity and the activity of each antioxidant enzyme in the same plant under stress were positively correlated. In our study, the changes in SOD and MDA activities in control plants and silenced plants under salt stress were smaller than the changes in POD activity, but they still decreased (SOD) or increased (MDA) overall, implying that the activities of POD, SOD, CAT and MDA changed in different degrees under certain abiotic stress conditions .
Comments 7: The English language requires polish. For instance,
(1) The Abstract should concise, 11 stress marker genes for GhANN4-silenced plants , but none description of the marker genes for GhANN11?
(2) Line 248 “different stress” should be “different stresses”
Response 7: Thank you for the helpful reminder. We have sent the manuscript to native English speakers for polishing.
(1) Thank you for the helpful reminder. We have included information on marker genes for GhANN11 in the abstract. Please see lines 29 to 30. Due to the abstract word limit, we have omitted the relevant results after low-temperature treatment and highlighted the results of high-temperature treatment after GhANN11 We found that the silencing of GhANN11 improved the resistance of upland cotton plants to high temperature.
(2) Thank you for the helpful reminder. The phrase “different stress” was changed to “different stresses” on line 264.
Reviewer 2 Report
Comments and Suggestions for Authors
Dear Authors,
You have done a commendable job on this work. However, I have some concerns for improvement.
1. Line 39 rephase the lines distinguishing classes of ANN. Class A B vertebrate is confusing.
2. Need a table for localization of the genes.
3. Fig 6 is missing statistical evaluation.
4. For Title 4.8, mention it as stress treatment. It is not the method of evaluation.
5. Need details of POD SOD CAT and MDA analysis. Just citing is not sufficiant here. Atleast a few details will help other readers.
6. Phylogenetic tree needs to be enlarged. It is too tiny to be visible.
7. Please check althrough out the paper the usage of ANN and annexins. Keep it similar.
8. Can a hypothesis be put up for the selected gene. How is it helping the plant. Since you have tested some stress related genes. Please put this up in the discussion.
9. Was there any change in the photosysnthetic parameters?
10. For gossypium fibre quality is very important? Can the authors comment
on this aspect?
11. What does soaking in the NaCl and PEG mean? Please rephrase this part.
Author Response
Thank you for the informative comments and helpful suggestions from you . We have studied the comments carefully and have made revisions. All the comments were valuable and provided important guidance for our research. The manuscript has been polished by English-speaking experts. The revisions made according to the comments from the reviewers have further improved our manuscript. We carefully revised the manuscript accordingly, and the changes are marked in yellow in the text. The responses are as follows:
Comments 1: Line 39 rephase the lines distinguishing classes of ANN. Class A B vertebrate is confusing.
Response 1: We apologize for the confusion. The phrase “vertebrate ANNs belong to class A; vertebrate ANNs are classified as class B” was changed to “vertebrate ANNs belong to class A; invertebrate ANNs are classified as class B” on lines 40 to 41.
Comments 2: Need a table for localization of the genes.
Response 2: Thank you for the helpful suggestion. Localization was previously in Table S1, but to make the information easier to read, we added the upland cotton localization information to Table 1. Please see line 173.
Comments 3: Fig 6 is missing statistical evaluation.
Response 3: Thank you for this constructive suggestion. We have completed the statistical evaluation in Figure 6. Please see line 282.
Comments 4: For Title 4.8, mention it as stress treatment. It is not the method of evaluation.
Response 4: Thank you for the helpful reminder. The phrase “Evaluation of salt, drought, cold and heat resistance” was changed to “Salt, Drought, Cold and Heat Resistance Stress Treatment” on line 593.
Comments 5: Need details of POD SOD CAT and MDA analysis. Just citing is not sufficiant here. At least a few details will help other readers.
Response 5: Thank you for the helpful suggestion. We added some details about the methods for determining POD, SOD, CAT and MDA activity on lines 601 to 612.
Comments 6: Phylogenetic tree needs to be enlarged. It is too tiny to be visible.
Response 6: Thank you for the helpful reminder. We have enlarged the phylogenetic tree. Please see line 184.
Comments 7: Please check althrough out the paper the usage of ANN and annexins. Keep it similar.
Response 7: Thank you for the helpful reminder. We have checked the use of “ANN” and “annexins” throughout the text to maintain consistency. ANN is short for Annexin, and in our document we use the full Annexin when we first use it and the short ANN when we use it again.
Comments 8: Can a hypothesis be put up for the selected gene. How is it helping the plant. Since you have tested some stress related genes. Please put this up in the discussion.
Response 8: Thank you for the helpful suggestion. We have added “On the basis of the above results, we speculated that GhANN4 positively regulates the tolerance of upland cotton to salt and drought and that GhANN11 positively regulates the tolerance of upland cotton to low temperature and negatively regulates the tolerance of upland cotton to high temperature.” to the discussion section on lines 503 to 506.
Comments 9: Was there any change in the photosysnthetic parameters?
Response 9: This is a good question. We have measured the data of the relevant photosynthetic parameters in the laboratory. Nevertheless, the data is not stable. According to our current analysis, GhANN11 is more noteworthy than GhANN4 in regulating upland cotton response to abiotic stress. Other members of our laboratory are conducting the determination of relevant field physiological data (including photosynthetic parameters) on the GhANN11 overexpressed lines and gene-edited mutant transgenic lines obtained. We will continue to share our research in this respect in the future.
Comments 10: For gossypium fibre quality is very important? Can the authors comment on this aspect?
Response 10: This is a good question. The GhAnx1 (GhANN25), GhAnn2 and GhFAnnxA (GhANN18) genes are highly expressed in cotton fibers and participate in the formation and elongation of cotton fiber cells, which is important for the elongation and quality of cotton fibers. We found that these genes clustered together with GhANN4, GhANN20, GhANN21, GhANN22, GhANN23 and GhANN24 in clade A and clade F. GhANN4 was subjected to stress treatment in our study; therefore, we speculate that GhANN4 may also be involved in the formation and elongation of fibers. However, GhANN11 clustered in clade C, and we found that GhANN11 and cotton annexin fiber-related genes have different structures; therefore, we speculate that they have different functions.
Comments 11: What does soaking in the NaCl and PEG mean? Please rephrase this part.
Response 11: Thank you for the helpful reminder. The phrase “For salt stress tolerance evaluation, plants were soaked with 400 mM NaCl for 35 d. Drought tolerance was evaluated by soaking the plants with 15% PEG for 35 d” was changed to “The roots of both control and target GhANN4-silenced plants were irrigated with 400 mM NaCl as salt stress up to 35 d. The roots of both control and target GhANN4-silenced plants were irrigated with 15% PEG6000 as drought stress up to 35 d. ” on lines 595 to 597.
Round 2
Reviewer 2 Report
Comments and Suggestions for Authors
Dear Authors,
Thank you for resubmitting the MS. The work and the text has been improved to a great extent.
A last query in the bar graphs relative expreddion is mentioned. Is it a correct word? Please revisit.
Author Response
Thank you for the informative comments and helpful suggestions from you . We have studied the comments carefully and have made revisions. All the comments were valuable and provided important guidance for our research. The manuscript has been polished by English-speaking experts. The revisions made according to the comments from the reviewers have further improved our manuscript. We carefully revised the manuscript accordingly, and the changes are marked in yellow in the text. The responses are as follows:
Question: A last query in the bar graphs relative expreddion is mentioned. Is it a correct word? Please revisit.
Answer: Thank you for the helpful reminder. We have changed "relative expreddion" to "relative expression" in the Figure 7F, 8F 9F and 10F.